# Joint Line Segmentation and Transcription for End-to-End Handwritten Paragraph Recognition

**Théodore Bluche**
A2iA SAS
39 rue de la Bienfaisance
75008 Paris
tb@a2ia.com

## Abstract

Offline handwriting recognition systems require cropped text line images for both training and recognition. On the one hand, the annotation of position and transcript at line level is costly to obtain. On the other hand, automatic line segmentation algorithms are prone to errors, compromising the subsequent recognition. In this paper, we propose a modification of the popular and efficient Multi-Dimensional Long Short-Term Memory Recurrent Neural Networks (MDLSTM-RNNs) to enable end-to-end processing of handwritten paragraphs. More particularly, we replace the collapse layer transforming the two-dimensional representation into a sequence of predictions by a recurrent version which can select one line at a time. In the proposed model, a neural network performs a kind of implicit line segmentation by computing attention weights on the image representation. The experiments on paragraphs of Rimes and IAM databases yield results that are competitive with those of networks trained at line level, and constitute a significant step towards end-to-end transcription of full documents.

## 1   Introduction

Offline handwriting recognition consists in recognizing a sequence of characters in an image of handwritten text. Unlike printed texts, images of handwriting are difficult to segment into characters. Early methods tried to compute segmentation hypotheses for characters, for example by performing a heuristic over-segmentation, followed by a scoring of groups of segments (e.g. in [4]). In the nineties, this kind of approach was progressively replaced by segmentation-free methods, where a whole word image is fed to a system providing a sequence of scores. A lexicon constrains a decoding step, allowing to retrieve the character sequence. Some examples are the sliding window approach [25], in which features are extracted from vertical frames of the line image, or space-displacement neural networks [4]. In the last decade, word segmentations were abandoned in favor of complete text line recognition with statistical language models [10].

Nowadays, the state of the art handwriting recognition systems are Multi-Dimensional Long Short-Term Memory Recurrent Neural Networks (MDLSTM-RNNs [18]), which consider the whole image, alternating MDLSTM layers and convolutional layers. The transformation of the 2D structure into a sequence is computed by a simple collapse layer summing the activations along the vertical axis. Connectionist Temporal Classification (CTC [17]) allows to train the network to both align and recognize sequences of characters. These models have become very popular and won the recent evaluations of handwriting recognition [9, 34, 37].

However, current models still need segmented text lines, and full document processing pipelines should include automatic line segmentation algorithms. Although the segmentation of documents into lines is assumed in most descriptions of handwriting recognition systems, several papers or

surveys state that it is a crucial step for handwriting text recognition systems [8, 28]. The need of line segmentation to train the recognition system has also motivated several efforts to map a paragraph-level or page-level transcript to line positions in the image (e.g. recently [7, 16]).

Handwriting recognition systems evolved from character to word segmentation, and to complete line processing nowadays. The performance has always improved by making less segmentation hypotheses. In this paper, we pursue this traditional tendency. We propose a model for multi-line recognition based on the popular MDLSTM-RNNs, augmented with an attention mechanism inspired from the recent models for machine translation [3], image caption generation [38], or speech recognition [11, 12]. In the proposed model, the *"collapse"* layer is modified with an attention network, providing weights to modulate the importance given at different positions in the input. By iteratively applying this layer to a paragraph image, the network can transcribe each text line in turn, enabling a purely segmentation-free recognition of full paragraphs.

We carried out experiments on two public datasets of handwritten paragraphs: Rimes and IAM. We report results that are competitive with the state-of-the-art systems, which use the ground-truth line segmentation. The remaining of this paper is organized as follows. Section 2 presents methods related to the one presented here, in terms of the tackled problem and modeling choices. In Section 3, we introduce the baseline model: MDLSTM-RNNs. We expose in Section 4 the proposed modification, and we give the details of the system. Experimental results are reported in Section 5, and followed by a short discussion in Section 6, in which we explain how the system could be improved, and present the challenge of generalizing it to complete documents.

## 2   Related Work

Our work is clearly related to MDLSTM-RNNs [18], which we improve by replacing the simple collapse layer by a more elaborated mechanism, itself made of MDLSTM layers. The model we propose iteratively performs an implicit line segmentation at the level of intermediate representations.

Classical text line segmentation algorithms are mostly based on image processing techniques and heuristics. However, some methods were devised using statistical models and machine learning techniques such as hidden Markov models [8], conditional random fields [21], or neural networks [24, 31, 32]. In our model, the line segmentation is performed implicitly and integrated in the neural network. The intermediate features are shared by the transcription and the segmentation models, and they are jointly trained to minimize the transcription error.

Recently, many "attention-based" models were proposed to iteratively select in an encoded signal the relevant parts to make the next prediction. This paradigm, already suggested by Fukushima in 1987 [15], was successfully applied to various problems such as machine translation [3], image caption generation [38], speech recognition [11, 12], or cropped words in scene text [27]. Attention mechanisms were also parts of systems that can generate or recognize small pieces of handwriting (e.g. a few digits with DRAW [20] or RAM [2], or short online handwritten sequences [19]). Our system is designed to handle long sequences and multiple lines.

In the field of computer vision, and particularly object detection and recognition, many neural architectures were proposed to both locate and recognize the objects, such as OverFeat [35] or spatial transformer networks (STN [22]). In a sense, our model is quite related to the DenseCap model for image captioning [23], itself similar to STNs. However, we do not aim at explicitly predicting line positions, and STNs are not as good with a large amount of small objects.

We recently proposed an attention-based model to transcribe full paragraphs of handwritten text, which predicts each character in turn [6]. Outputting one token at a time turns out to be prohibitive in terms of memory and time consumption for full paragraphs, which typically contain about hundreds of characters. In the proposed system, the encoded image is not summarized as a single vector at each timestep, but as a sequence of vectors representing full text lines. It represents a huge speedup, and a comeback to the original MDLSTM-RNN architecture, in which the collapse layer is augmented with an MDLSTM attention network similar to the one presented in [6].

## 3   Handwriting Recognition with MDLSTM and CTC

MDLSTM-RNNs [18] were first introduced in the context of handwriting recognition. The Multi-

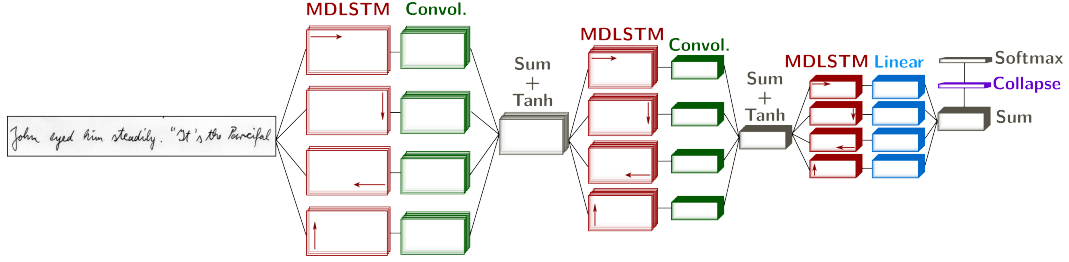

Figure 1: MDLSTM-RNN architecture for handwriting recognition. LSTM layers in four scanning directions are followed by convolutions. The feature maps of the top layer are are summed in the vertical dimension, and character predictions are obtained after a softmax normalization.

Dimensional Long Short-Term Memory layers scan the input in the four possible directions. The LSTM cell inner state and output are computed from the states and outputs of previous positions in the considered horizontal and vertical directions. Each MDLSTM layer is followed by a convolutional layer. At the top of this network, there is one feature map for each character. These maps are collapsed into a sequence of prediction vectors, normalized with a softmax activation. The whole architecture is depicted in Figure 1. The Connectionist Temporal Classification (CTC [17]) algorithm, which considers all possible labellings of the sequence, may be applied to train the network to recognize text lines.

The 2D to 1D conversion happens in the collapsing layer, which computes a simple aggregation of the feature maps into vector sequences, i.e. maps of height 1. This is achieved by a simple sum across the vertical dimension:

$$z_i = \sum_{j=1}^{H} a_{ij} \tag{1}$$

where $z_i$ is the $i$-th output vector and $a_{ij}$ is the input feature vector at coordinates $(i, j)$. All the information in the vertical dimension is reduced to a single vector, regardless of its position in the feature maps, preventing the recognition of multiple lines within this framework.

## 4 An Iterative Weighted Collapse for End-to-End Handwriting Recognition

In this paper, we replace the sum of Eqn. 1 by a weighted sum, in order to focus on a specific part of the input. The weighted collapse is defined as follows:

$$z_i^{(t)} = \sum_{j=1}^{H} \omega_{ij}^{(t)} a_{ij} \tag{2}$$

where $\omega_{ij}^{(t)}$ are scalar weights between 0 and 1, computed at every time $t$ for each position $(i, j)$. The weights are provided by a recurrent neural network, illustrated in Figure 2, enabling the recognition of a text line at each timestep.

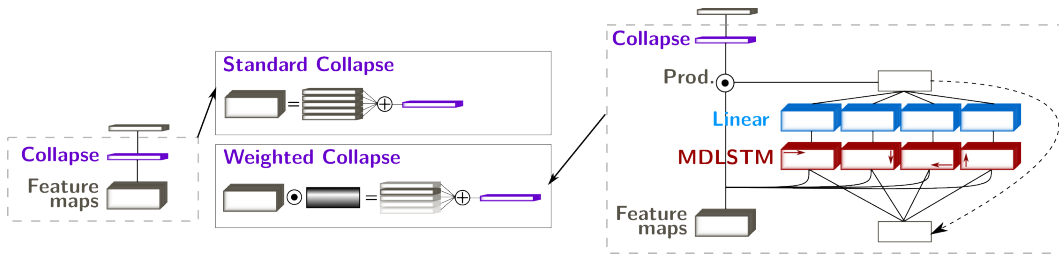

Figure 2: Proposed modification of the collapse layer. While the standard collapse (left, top) computes a simple sum, the weighted collapse (right, bottom) includes a neural network to predict the weights of a weighted sum.

This collapse, weighted with a neural network, may be interpreted as the "attention" module of an attention-based neural network similar to those of [3, 38]. This mechanism is differentiable and can be trained with backpropagation. The complete architecture may be described as follows.

An **encoder** extracts feature maps from the input image $\mathcal{I}$:

$$\mathbf{a} = (a_{ij})_{(i,j) \in [1,W] \times [1,H]} = Encoder(\mathcal{I}) \tag{3}$$

where $(i,j)$ are coordinates in the feature maps. In this work, the $Encoder$ module is an MDLSTM network with same architecture as the model presented in Section 3.

A **weighted collapse** provides a view of the encoded image at each timestep in the form of a weighted sum of feature vector sequences. The attention network computes a score for the feature vectors at every position:

$$\alpha_{ij}^{(t)} = Attention(\mathbf{a}, \omega^{(t-1)}) \tag{4}$$

We refer to $\omega^{(t)} = \{\omega_{ij}^{(t)}\}_{(1 \leq i \leq W, \ 1 \leq j \leq H)}$ as the attention map at time $t$, which computation depends not only on the encoded image, but also on the previous attention features. A softmax normalization is applied to each column:

$$\omega_{ij}^{(t)} = e^{\alpha_{ij}^{(t)}} / \sum_{j'} e^{\alpha_{ij'}^{(t)}} \tag{5}$$

In this work, the $Attention$ module is an MDLSTM network.

This module is applied several times to the features from the encoder. The output of the attention module at iteration $t$, computed with Eqn. 2, is a sequence of feature vectors $\mathbf{z}$, intended to represent a text line. Therefore, we may see this module as a soft line segmentation neural network. The advantages over the neural networks trained for line segmentation [13, 24, 32, 31] are that *(i)* it works on the same features as those used for the transcription (multi-task encoder) and *(ii)* it is trained to maximize the transcription accuracy (i.e. more closely related to the goal of handwriting recognition systems, and easily interpretable).

A **decoder** predicts a character sequence from the feature vectors:

$$\mathbf{y} = Decoder(\mathbf{z}) \tag{6}$$

where $\mathbf{z}$ is the concatenation of $z^{(1)}, z^{(2)}, \ldots, z^{(T)}$. Alternatively, the decoder may be applied to $z^{(i)}$s sub-sequences to get $y^{(i)}$s and $\mathbf{y}$ is the concatenation of $y^{(1)}, y^{(2)}, \ldots, y^{(T)}$.

In the standard MDLSTM architecture of Section 3, the decoder is a simple softmax. However, a Bidirectional LSTM (BLSTM) decoder could be applied to the collapsed representations. This is particularly interesting in the proposed model, as the BLSTM would potentially process the whole paragraph, allowing a modeling of dependencies across text lines.

This model can be trained with CTC. If the line breaks are known in the transcript, the CTC could be applied to the segments corresponding to each line prediction. Otherwise, one can directly apply CTC to the whole paragraph. In this work, we opted for that strategy, with a BLSTM decoder applied to the concatenation of all collapsing steps.

## 5   Experiments

### 5.1   Experimental Setup

We carried out the experiments on two public databases. The IAM database [29] is made of handwritten English texts copied from the LOB corpus. There are 747 documents (6,482 lines) in the training set, 116 documents (976 lines) in the validation set and 336 documents (2,915 lines) in the test set. The Rimes database [1] contains handwritten letters in French. The data consist of a training set of 1,500 paragraphs (11,333 lines), and a test set of 100 paragraphs (778 lines). We held out the last 100 paragraphs of the training set as a validation set.

The networks have the following architecture. The encoder first computes a 2x2 tiling of the input and alternate MDLSTM layers of 4, 20 and 100 units and 2x4 convolutions of 12 and 32 filters with no overlap. The last layer is a linear layer with 80 outputs for IAM and 102 for Rimes. The attention network is an MDLSTM network with 2x16 units in each direction followed by a linear

layer with one output, and a softmax on columns (Eqn. 5). The decoder is a BLSTM network with 256 units. Dropout is applied after each LSTM layer [33]. The networks are trained with RMSProp [36] with a base learning rate of 0.001 and mini-batches of 8 examples, to minimize the CTC loss over entire paragraphs. The measure of performance is the Character (or Word) Error Rate (CER%), corresponding to the edit distance between the recognition and ground-truth, normalized by the number of ground-truth characters.

## 5.2 Impact of the Decoder

In our model, the weighted collapse method is followed by a BLSTM decoder. In this experiment, we compare the baseline system (standard collapse followed by a softmax) with the proposed model. In order to dissociate the impact of the weighted collapse from that of the BLSTM decoder, we also trained an intermediate architecture with a BLSTM layer after the standard collapse, but still limited to text lines.

Table 1: Character Error Rates (%) of CTC-trained RNNs on 150 dpi images. The *Standard* models are trained on segmented lines. The *Attention* models are trained on paragraphs.

| Collapse | Decoder | IAM | Rimes |
|----------|---------|-----|-------|
| Standard | Softmax | 8.4 | 4.9 |
| Standard | BLSTM + Softmax | 7.5 | 4.8 |
| Attention | BLSTM + Softmax | 6.8 | 2.5 |

The character error rates (CER%) on the validation sets are reported in Table 1 for 150dpi images. We observe that the proposed model outperforms the baseline by a large margin (relative 20% improvement on IAM, 50% on Rimes), and that the gain may be attributed to both the BLSTM decoder, and the attention mechanism.

## 5.3 Impact of Line Segmentation

Our model performs an implicit line segmentation to transcribe paragraphs. The baseline considered in the previous section is somehow cheating, because it was evaluated on the ground-truth line segmentation. In this experiment, we add to the comparison the baseline models evaluated in a real scenario where they are applied to the result of an automatic line segmentation algorithm.

Table 2: Character Error Rates (%) of CTC-trained RNNs on ground-truth lines and automatic segmentation of paragraphs with different resolutions. The last column contains the error rate of the attention-based model presented in this work, without an explicit line segmentation.

| Database | Resolution | Line segmentation | | | | This work |
| | | GroundTruth | Projection | Shredding | Energy | |
|----------|------------|-------------|------------|-----------|--------|-----------|
| IAM | 150 dpi | 8.4 | 15.5 | 9.3 | 10.2 | 6.8 |
| | 300 dpi | 6.6 | 13.8 | 7.5 | 7.9 | 4.9 |
| Rimes | 150 dpi | 4.8 | 6.3 | 5.9 | 8.2 | 2.8 |
| | 300 dpi | 3.6 | 5.0 | 4.5 | 6.6 | 2.5 |

In Table 2, we report the CERs obtained with the ground-truth line positions, with three different segmentation algorithms, and with our end-to-end system, on the validation sets of both databases with different input resolutions. We see that applying the baseline networks on automatic segmentations increases the error rates, by an absolute 1% in the best case. We also observe that the models are better with higher resolutions.

Our models yield better performance than methods based on an explicit and automatic line segmentation, and comparable or better results than with ground-truth segmentation, even with a resolution divided by two. Two factors may explain why our model yields better results than the line recognition from ground-truth segmentation. First, the ground-truth line positions are bounding boxes that may include some parts of adjacent lines and include irrelevant data, whereas the attention model will focus on smaller areas. But the main reason is probably that the proposed model includes a BLSTM operating on the whole paragraph, which may capture linguistic dependencies across text lines.

In Figure 3, we display a visualisation of the implicit line segmentation computed by the network. Each color corresponds to one step of the iterative weighted collapse. On the images, the color represents the weights given by the attention network (the transparency encodes their intensity). The texts below are the predicted transcriptions, and chunks are colored according to the corresponding timestep of the attention mechanism.

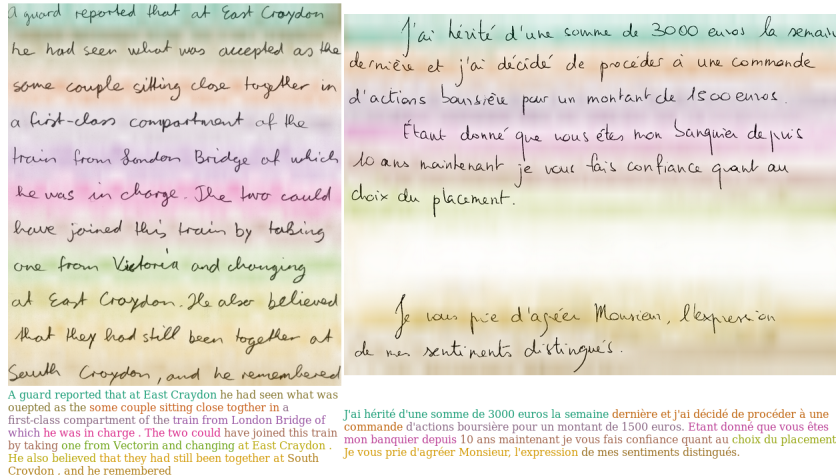

Figure 3: Transcription of full paragraphs of text and implicit line segmentation learnt by the network on IAM (left) and Rimes (right). Best viewed in color.

## 5.4 Comparison to Published Results

In this section, we also compute the word error rates (WER%) and evaluate our models on the test sets to compare the proposed approach to existing systems. For IAM, we applied a 3-gram language model with a lexicon of 50,000 words, trained on the LOB, Brown and Wellington corpora.[1] This language model has a perplexity of 298 and out-of-vocabulary rate of 4.3% on the validation set (329 and 3.7% on the test set).

The results are presented in Table 3 for different input resolutions. When comparing the error rates, it is important to note that all systems in the literature used an explicit (ground-truth) line segmentation and a language model. [14, 26, 30] used a hybrid character/word language model to tackle the issue of out-of-vocabulary words. Moreover, all systems except [30, 33] carefully pre-processed the line image (e.g. corrected the slant or skew, normalized the height, ...), whereas we just normalized the pixel values to zero mean and unit variance. Finally, [5] is a combination of four systems.

Table 3: Final results on Rimes and IAM databases

|  |  | Rimes | | IAM | |
|---|---|---|---|---|---|
|  |  | WER% | CER% | WER% | CER% |
| **150 dpi** | no language model | 13.6 | 3.2 | 29.5 | 10.1 |
|  | with language model |  |  | 16.6 | 6.5 |
| **300 dpi** | no language model | 12.6 | **2.9** | 24.6 | 7.9 |
|  | with language model |  |  | 16.4 | 5.5 |
|  | Bluche, 2015 [5] | **11.2** | 3.5 | **10.9** | **4.4** |
|  | Doetsch et al., 2014 [14] | 12.9 | 4.3 | 12.2 | 4.7 |
|  | Kozielski et al. 2013 [26] | 13.7 | 4.6 | 13.3 | 5.1 |
|  | Pham et al., 2014 [33] | 12.3 | 3.3 | 13.6 | 5.1 |
|  | Messina & Kermorvant, 2014 [30] | 13.3 | - | 19.1 | - |

On Rimes, the system applied to 150 dpi images already outperforms the state of the art in CER%, while being competitive in terms of WER%. The system for 300 dpi images is comparable to the best single system [33] in WER% with a significantly better CER%.

On IAM, the language model turned out to be quite important, probably because there is more variability in the language.[2] On 150 dpi images, the results are not too far from the state of the art results. The WER% does not improve much on 300 dpi images, but we get a lower CER%. When analysing the errors, we noticed that there is a lot of punctuation in IAM, which was often missed by the attention mechanism. It may happen because punctuation marks are significantly smaller than characters. With the attention-based collapse and the weighted sum, they will be more easily missed than with the standard collapse, which gives the same weight to all vertical positions.

## 6  Discussion

Table 4: Comparison of decoding times of different methods: using ground-truth line information, with explicit segmentation, with the attention-based method of [6] and with the system presented in this paper.

| Method | | Processing time (s) |
|---|---|---|
| GroundTruth | (crop+reco) | $0.21 \pm 0.07$ |
| Shredding | (segment+crop+reco) | $0.78 \pm 0.26$ |
| Scan, Attend and Read [6] | (reco) | $21.2 \pm 5.6$ |
| This Work | (reco) | $0.62 \pm 0.14$ |

The proposed model can transcribe complete paragraphs without segmentation and is orders of magnitude faster that the model of [6] (cf. Table 4). However, the mechanism cannot handle arbitrary reading orders. Rather, it implements a sort of implicit line segmentation. In the current implementation, the iterative collapse runs for a fixed number of timesteps. Yet, the model can handle a variable number of text lines, and, interestingly, the focus is put on interlines in the additional steps. A more elegant solution should include the prediction of a binary variable indicating when to stop reading.

Our method was applied to paragraph images, so a document layout analysis is required to detect those paragraphs before applying the model. Naturally, the next step should be the transcription of complex documents without an explicit or assumed paragraph extraction. The limitation to paragraphs is inherent to this system. Indeed, the weighted collapse always outputs sequences corresponding to the whole width of the encoded image, which, in paragraphs, may correspond to text lines. In order to switch to full documents, several issues arise. On the one hand, the size of the lines is determined by the size of the text block. Thus a method should be devised to only select a smaller part of the feature maps, representing only the considered text line. This is not possible in the presented framework. A potential solution could come from spatial transformer networks [22], performing a differentiable crop. On the other hand, training will in practice become more difficult, not only because of the complexity of the task, but also because the reading order of text blocks in complex documents cannot be exactly inferred in many cases (even defining arbitrary rules may be tricky).

## 7  Conclusion

We have presented a model to transcribe full paragraphs of handwritten texts without an explicit line segmentation. Contrary to classical methods relying on a two-step process (segment, then recognize), our system directly considers the paragraph image without an elaborated pre-processing, and outputs the complete transcription. We proposed a simple modification of the collapse layer in the standard MDLSTM architecture to iteratively focus on single text lines. This implicit line segmentation is learnt with backpropagation along with the rest of the network to minimize the CTC error at the paragraph level. We reported error rates comparable to the state of the art on two public databases. After switching from explicit to implicit character, then word segmentation for handwriting recognition, we showed that line segmentation can also be learnt inside the transcription model. The next step towards end-to-end handwriting recognition is now at the full page level.

## Footnotes

[1] The parts of the LOB corpus used in the validation and evaluation sets were removed.

[2] A simple language model yields a perplexity of 18 on Rimes [5].

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
