[Reviews · NeurIPS 2016]

Reviewer 1

Summary

The authors have developed an end-to-end system for parsing hand written text from paragraph (multiple horizontal lines) images. They achieve this by adding an attention mechanism to the single-line parser of Graves et al. [18] and then decoding the text using Bidirectional-LSTMs (instead of softmax as used originally by Graves et al). They evaluate recognition performance on the Rimes and IAM datasets. The performance is comparable to the state-of-the-art methods on Rimes, but lags behind on IAM.

Qualitative Assessment

This is a very nice application of guided attention over an image for identifying a sequence of handwritten text lines (as the authors correctly point out, this builds on the approach of Xu et al [38] and others, who have used region attention over an image). On the exposition, the paper could be written to be more self-contained. It is difficult to understand the model as presented in the paper without first referring to Graves et al. [18]. It would help to improve section 3 if the MD-LSTM model is explained in more detail (as in [18]). There are a few peculiar results and statements that should be explained by the authors: - why does the proposed approach give better performance than the methods that use ground truth line labelling (in table 2 & section 5.3)? Is it because the line breaks in the ground truth are used to re-initialize the decoder RNN, but this re-initialization does not happen in the character sequence of the attention mechanism? More details on the authors' implementation of Graves et al should be provided to answer this question. - line 142, the authors say their method is 20-30x faster than the character decoder method of Bluche et al. [6], but do not provide the actual time for comparison. - In terms of results, the performance is superior to the state of the art (for one setting) for CER% on the Rimes dataset, but substantially worse on the IAM dataset, probably due to lack of training data. A simple experiment to test this would be to first train the model on the test+training data of the other dataset and then finetune on the IAM training data. I suggest the authors add this experiment. - line 213, why does the attention mechanism miss punctuation?

Confidence in this Review

2-Confident (read it all; understood it all reasonably well)


Reviewer 2

Summary

The paper provides a useful approach to jointly segmenting and transcribing handwritten paragraphs to text. The authors use MDLSTMs to learn 1) the encoding of the input and 2) an attention network that effectively segments each line of text. A Bidirectional-LSTM decoder is trained to output the text corresponding to the concatenation of the encoded sequences. A CTC loss is minimized over the entire paragraph.

Qualitative Assessment

The paper is a useful progression of previous work extending beyond models that need line segmentation. The model has real-world applicability. Some feedback on how the paper could have been stronger: It would have been useful to have a baseline with just using Convolutional or Affine [5] layers since these would be much simpler to train compared to using MDLSTMs. The complexity of the model will increase if another level of MDLSTMs with CTCs are used to find the paragraph boundaries. While the authors do find improvements over their own baseline and improved performance on the Rimes benchmark, the performance on the IAM benchmark was not an improvement over previous results.

Confidence in this Review

2-Confident (read it all; understood it all reasonably well)


Reviewer 3

Summary

This paper addresses the problem of end-to-end handwriting recognition at paragraph-level. Conventional approaches consist of two steps: Line segmentation and line recognition. This work unifies the two steps using an attention-based model. The experimental results show that the proposed approach outperforms the conventional two-step approaches even if the conventional approaches use the ground-truth line segmentation. It also shows that the proposed approach provides good results compared to state-of-the-art line-level methods reported on Rimes and IAM datasets even though the proposed approach does not use sophisticated pre-processing methods which were used in the previous methods.

Qualitative Assessment

The state-of-the-art handwriting recognition has been taking the two-step approach for decades and researchers have been focusing on improving the individual components. End-to-end line-segmentation-free algorithms have been desired but it has been a challenging problem. However, recent advances in deep learning have opened the door to build such systems. This work comes in the context. This work proposes to unify the two steps by using an attention-based model. The attention mechanism is used to do the line segmentation implicitly in the model. The formulation is very natural and makes sense. The experimental results show the effectiveness of the approach. Given the current trends of the related areas, applying attention-based models to the end-to-end paragraph-level handwriting recognition is something expected to be done. As far as I know, this is the first work which actually does it. The paper is well written and easy to understand. Comments: It needs some explanation about why the attention-based model is better than the line-based model with the ground-truth line segmentation. Reporting computational time will be valuable.

Confidence in this Review

3-Expert (read the paper in detail, know the area, quite certain of my opinion)


Reviewer 4

Summary

This work introduces a modification to the collapse layer of standard MD-LSTMS used to do handwriting recognition introduced in [18]. The work is also closely related to [6] that also performs the same task -- decoding at a character level per time step. The modification is to convert the global average pooling into a weighted average pooling (eq. 2). Further, these weights are learned using a recurrent attention layer (MD-LSTM) that takes in both the previous weights and the encoded feature vector. A CTC loss is used to directly compute loss for the transcription without needing character or line-level alignment. Moreover, the attention mechanism can be interpreted as a “soft” line segmentation. The work provides comparison with other methods including [6] and obtains better / comparable performance.

Qualitative Assessment

From my understanding, the major change from [6] to this paper is that at each time step an entire line is decoded as against a single character. Although this results in improved speeds and better performance, I am not completely convinced that there is enough technical novelty. (Please correct me in the rebuttal if I am missing something and if so, I will be happy to increase my rating) Also, it would have been interesting to visualize attentions when number of lines in input image is greater than the number of time steps the algorithm is set to run -- as it is mentioned that the method still fares better in such a scenario.

Confidence in this Review

2-Confident (read it all; understood it all reasonably well)


Reviewer 5

Summary

The authors investigate the introduction of an attention mechanism into an offline handwriting recognition system pointed at unsegmented paragraphs of handwritten text. MD-LSTMs are used for feature extraction, attention computation, and (in some cases) decoding. Interestingly, learned attention is shown to outperform ground truth segmentation in the authors' system. Character error rates on standard datasets appear to be quite impressive. While the system's word error rates do not break state of the art, they are quite competitive despite ignoring both ground truth segmentation and employing a relatively simplistic language model.

Qualitative Assessment

Positives: -The idea of applying technique to generalize single-line handwriting recognition to multi-line is both an elegant idea and a natural next step. It is unsurprising that this is one of at least two works (mentioned by the authors) attempting similar techniques (not that this is relevant, but I consider this to be the more mature/developed work of the two). -It is impressive (as is often the case these days) that such a system works at all, let alone that it works as well as it does. -The technical exposition provided by the authors is sufficient, and is generally easy to follow. The colorized attention figure is particularly gratifying. Areas for improvement: -The comparison to established benchmarks raises more questions than it provides answers. As the authors note, this comparison is apples-to-oranges. The reference models exploit more complex language models and make use of ground truth segmentation. The latter is unavoidable due to the fundamental nature of the problem considered, but the gap in language models is quite unnecessary. Discussion of this comparison also feels underdeveloped and unenlightening, and I am confident the authors could do better. -The authors do not adequately discuss the significance/origin of the learned attention outperforming the ground truth. Is it because of noise in the annotations? Leveraging information contained in neighboring lines? This is particularly important, as the models compared to in Sec 5.4 all make use of this ground truth. -Finally, while the paper is well organized and easy to follow, the quality of the writing leaves something to be desired. There are far too many typos, grammatical errors, and shifting tenses. Furthermore, particularly when comparing against other work, the occasionally competitive tone undermines the authors' credibility. Their results stand up quite well, even in a neutral light.

Confidence in this Review

2-Confident (read it all; understood it all reasonably well)


Reviewer 6

Summary

This paper proposes a joint segmentation and transcription for handwritten paragraph recognition. The paper builds on past work on using multidimensional LSTM-RNNs for this purpose. Such a model employs the connectionist temporal classification (CTC) to train the network. This paper builds on reference [6] by employing a weighted 'collapsing' step where a weighted sum of the feature vector sequence is formed before the final step. This weighted collapse induces an attentional mechanism into the recognition step and improves the performance. The final step uses a bidirectional LSTM decoder instead of a softmax. In doing so, the decoder outputs entire text lines instead of individual characters. This provides a speed-up of 20X to 30X probably in terms of the number of iterations. The proposed method has improved recognition accuracy than past approaches.

Qualitative Assessment

The contributions of this paper is in suggesting a weighted collapse layer (provides attentional mechanism) and the use of a BLSTM decoder (to output sentences) for handwriting recognition. These are reasonable next steps that augment earlier work but seem to lack an element of surprise. The improvement in accuracy is made very clear in the results and these are quite good. However, the other claim, which is regarding the speed-up, is not quantified well. A major issue with this manuscript are the large number of typos. Hopefully, the authors can correct these if the manuscript is accepted for publication.

Confidence in this Review

2-Confident (read it all; understood it all reasonably well)